# OpenReview forum: "From Prompts to Perception: Auditing Stereotypes in Multimodal AI"
_TMLR — Rejected by TMLR_

### Review · Reviewer_xsPt · 2026-01-02

**Summary Of Contributions:**

The paper introduces a model-agnostic auditing framework designed to quantify how stereotypes propagate through combined text-to-image (T2I) and multimodal large language model (MLLM) pipelines. Its primary contributions include:
* Novel Metrics: The development of five metrics (TDS, WTD, SDI, OM, and MCS) to measure physical trait shifts, label dominance, and model stability under image degradation.
* Systematic Evaluation: A comprehensive audit of four T2I models and five MLLMs using a diverse dataset of 15,440 images across seven nationalities and five gender terms.
* Analysis of Bias Propagation: Identification of specific stereotypical patterns, such as the drift of "gender-neutral" prompts toward "man" and the over-reliance on nationality-specific visual cues (e.g., associating Indian individuals with "yoga" or "engineering").
* Stability Assessment: An investigation into how natural image corruptions (noise, blurring) affect the consistency of MLLM predictions, revealing that even mild degradation can lead to significant label changes.

**Audience:**

Yes

**Audience Explanation:**

The paper addresses several high-interest areas for the TMLR community:


* Fairness and Ethics: As MLLMs become more pervasive, understanding their societal impact and inherent biases is a critical area of research.



* Framework Utility: The model-agnostic nature of the auditing framework makes it a valuable tool for researchers and developers looking to evaluate their own models.




* Robustness: The analysis of how image corruption influences bias (RQ3) connects fairness research with model robustness, another core area of machine learning.

**Broader Impact Concerns:**

The paper proactively addresses broader impact by:


* Identifying Harmful Labels: It notes that models occasionally assign highly problematic labels like "criminal" or "terrorist" to specific demographics, which has severe ethical implications for real-world deployment.



* No Intent to Offend: The authors explicitly state that their goal is to showcase model limitations rather than cause offense or disrespect.


* Ethical Responsibility: By highlighting how visual cues (clothing, facial attributes) trigger stereotypes, the paper serves as a warning for the "ethical and responsible deployment" of these technologies in sensitive areas like hiring or security.

**Claims And Evidence:**

Yes

**Claims Explanation:**

The authors provide a structured and transparent methodology to support their findings:


* Quantitative Rigor: Claims regarding nationality and gender bias are supported by statistically significant results from t-tests and Mann-Whitney U tests.



* Visual and Mathematical Clarity: The use of t-SNE visualizations, radar plots, and treemaps clearly illustrates the "clustering" of stereotypes and the distance of specific groups (like Nigerian) from a neutral baseline.




* Reproducibility: The authors define clear prompt templates and experimental setups, including the use of both open-source (SDXL, Flux) and closed-source (SORA) models, which strengthens the model-agnostic claim.



* Comparison to Baselines: The inclusion of a "neutral baseline" (images generated without nationality tags) allows for a clear measurement of "deviation," providing a concrete point of comparison for identifying bias.

**Requested Changes:**

While the paper is technically sound, the following changes would improve its quality:
* Expansion of Demographic Attributes: While the paper acknowledges its focus is primarily on nationality, further discussion on how this framework could be extended to other intersectional attributes (e.g., age or disability) would be beneficial.
* In-Depth Mitigation Strategies: The authors mention that their framework facilitates "targeted mitigation strategies," but providing a few concrete examples of how these metrics could be used during the training or fine-tuning phase to reduce bias would add significant value.

---

> ### Author Response · Authors · 2026-01-19
> **Response for Reviewer xsPt**
>
> We sincerely thank Reviewer xsPt for their constructive feedback and time. As for the concerns, we have addressed them as follows -
>
> **1. Extension to other intersectional attributes:** While our experiments focus on nationality and gender, the framework is intentionally attribute-agnostic. The same prompt construction, neutral baselines, and metrics can be directly applied to other demographic or intersectional attributes such as age, disability, or socioeconomic cues, without methodological changes. We have some age-coded gender terms (boy/man and girl/woman) that indicate that age-related associations are relevant and very much present. In the UTKFaceDataset as well, we observed that younger individuals were often assigned complex occupations, which might be seen as indicative of a lack of reasoning. When querying the dataset using the LLaVA-OneVision model, we got the following results (the dataset provided age for each sample, we have clubbed the ages to analyze a range):
>
> | Age Group | Age Range | Total Count of Samples in the Range | Frequent Occupation Labels (Open-Form) |
> |---|---|---|---|
> | Kids | 0-16 | 4110 | Photographer (1794), parent (222), actor(195) |
> | Young Adult | 17-26 | 3820 | Actress (947), Actor (518), Photographer (464) |
> | Adult | 27-45 | 10025 | Actor (1516), Actress (1495), Athlete (710) |
> | Middle-Age | 45-61 | 3671 | Teacher (543), Actor (535), Actress (256) |
> | Old Age | 61-120 | 2448 | Teacher (262), Actress (260), Retired (215) |
>
> The results clearly indicate that over-reliance persists across all ranges, with actor, actress, photographer, and teacher among the most frequently assigned occupation terms. However, we can also see the effect of age group: occupations such as retired are assigned to the old age group, and Athlete is assigned to the young adult and adult groups. Parent is often assigned to the kids group, which can be interpreted in multiple ways. This is the next step in our analysis (as future work), analyzing the model’s reasoning behind these assignments. We have not included these results in the revised paper due to the lack of a formalized framework for the age dimension. But we do think that age can be a revealing dimension for different stereotypes.
>
> **2. Concrete examples of mitigation strategies:** Although our work focuses on auditing rather than proposing new mitigation algorithms, the metrics are designed to be actionable. A simple, actionable application would be to extend traits/attributes to the proposed framework as required and to use the proposed metrics as a regularization signal during fine-tuning. They can also be used to penalize over-reliance on specific labels or used as a post-deployment monitor to identify and discourage dominance of specific harmful labels.
>
> Both of these concerns have been addressed in the main paper in the Limitations and Conclusion sections, respectively.

---

### Review · Reviewer_aBQz · 2026-01-03

**Summary Of Contributions:**

- This paper presents an auditing framework that evaluates the propagation of stereotypes in T2I and MLLM pipelines, experimenting with four T2I models and five MLLMs across seven nationality and five gender groups.
- Two quantitative metrics for T2I evaluation are introduced, i.e., Trait Deviation Score and Weighted Trait Divergence, which utilize CLIP embeddings to measure how generated images for specific demographics deviate from a neutral baseline in terms of physical traits.
- To assess MLLM bias, the framework employs Stereotype Dominance Index and Overlap Matrix to analyze the frequency and overlap of predicted attributes, revealing an over-reliance on specific labels and categories.
- The framework further examines the stability of these biases through natural image corruptions, demonstrating that mild visual degradations lead to significant shifts in model predictions and label consistency.
- On the positive side, this paper addresses a critical problem with high societal impact by auditing the end-to-end propagation of bias from image generation to multimodal interpretation, empirically analyzing how mainstream models may implicitly enforce certain norms and stereotypes.
- However, I believe the experimental framework contains several nontrivial loopholes regarding the configuration, quantification, and assessment. Please refer to the next section for details.

**Additional Comments:**

- The footnote disclaimer regarding the lack of intention to cause offense (currently on Page 2) would be more appropriate and visible if placed on the first page.
- The use of treemaps to visualize categorical distributions is not immediately intuitive for readers. More legible alternatives could be considered, e.g., pie charts with shared legends, to improve the readability of the frequency analyses.

**Audience:**

Yes

**Audience Explanation:**

Investigating biases and stereotypes in frontier models is a critical and timely topic. A subset of the audience would likely find the empirical observations and proposed metrics in this paper relevant.

**Broader Impact Concerns:**

The draft has a broader impact statement, and I do not identify any major issues.

**Claims And Evidence:**

No

**Claims Explanation:**

- The experimental design relies on an arbitrary set of only seven nationalities (not supported by a clear rationale or provide sufficient demographical coverage) and a closed vocabulary of attributes, which significantly limits the generalizability of the findings. Compared to contemporaneous work like ViSAGe (ACL 2024), which covers orders of magnitude more groups, the scale of this study is insufficient to support broad claims about global stereotype auditing. Furthermore, the analysis primarily describes specific characteristics of these few categories rather than uncovering systemic, transferable patterns of bias.
- The reliance on CLIP embeddings to evaluate T2I bias could be methodologically weak, as it uses one potentially biased model to audit another without accounting for CLIP's own alignment issues. Additionally, despite the focus on distributional shifts in image generation, the framework omit standard, distribution-based evaluation metrics such as Frechet Distance, opting instead for custom metrics (SDI, OM) whose statistical robustness is not well-established against existing baselines.
- Moreover, it is questionable whether the proposed quantitative metrics for T2I evaluation are rigorous enough. These metrics measure deviation from a neutral baseline; however, if the neutral baseline is implicitly biased (e.g., defaulting to White/Western features, as evidenced by the low deviation for American images), then a model that faithfully generates diverse demographics will be penalized with a high bias score. Consequently, a perfect model under this framework would be one that generates generic, non-distinctive images regardless of the nationality prompt, effectively discouraging accurate demographic representation.
- The framework assesses MLLM bias by analyzing predictions of occupation or interest from generated images, but fails to isolate stereotypical reasoning from faithful visual recognition. MLLM predictions are heavily influenced by confounding visual factors introduced by the T2I model, such as attire, background context (e.g., airports), and image quality. It is unclear whether an MLLM labeling a person in religious garb as a priest is exhibiting a harmful stereotype or simply performing robust visual captioning, rendering the claim of auditing stereotype risk technically ambiguous.
- The analysis in RQ3, which claims that image corruption leads to meaningful changes in predictions, offers limited insight. It is a well-known phenomenon that deep learning models are susceptible to noise and perturbation. The paper does not convincingly demonstrate that this instability is specific to stereotypical logic versus general model fragility, nor does it prove that the _actionable tools_ claimed in the abstract provide any mechanism for mitigation beyond simple observation.

**Requested Changes:**

Please refer to the weaknesses section for a detailed breakdown of my concerns. I would appreciate detailed clarifications regarding the configuration, quantification, and assessment.

---

> ### Author Response · Authors · 2026-01-19
> **Response for Reviewer aBQz**
>
> We thank Reviewer aBQz for their constructive comments. We address the concerns raised with justifications as follows -
>
> **1. Limited demographic scope and comparison to ViSAGe:** We agree that our experimental scope is smaller than large-scale efforts such as ViSAGe. However, we have added three more nationalities (Chinese, Egyptian, and French) to showcase that the auditing framework is fully extendable with results updated throughout the revised paper. Our intent is to propose a diagnostic auditing framework that traces how stereotypes propagate across the multimodal pipeline from a T2I model to generation through an MLLM. Furthermore, the chosen ten nationalities, while not exhaustive, span multiple continents and cultural regions. This helps include different cultural symbols, clothing styles, facial attributes, and social cues to facilitate the investigation of various stereotypes. We utilized both overrepresented and underrepresented groups in the large-scale training datasets used for the models under consideration. As for the closed-vocabulary setting, we agree that it can limit generalizability (but showcase the model’s underlying patterns). We have therefore included an open-vocabulary setting.
>
> **2. Reliance on CLIP embeddings for auditing bias:** We agree that CLIP should not be treated as a fairness oracle. In our framework, CLIP is not used as a fairness ground truth. Further, we update trait alignment by including SigLIP-based image-text similarity alongside CLIP. The results are reported in Table 3 of the main paper, where TDS and WTD are computed for both models, and the average of the two is reported. In most cases, the values from the SigLIP model agree with those from the CLIP model.
>
> **3. Absence of standard distributional metrics, such as FID:** FID measures pixel-level realism and diversity, and is not designed to capture semantic or demographic stereotype shifts. Two image sets can have similar FID while encoding very different demographic associations.  With SDI and OM, we are quantifying semantic label dominance, operating at the level of attribute-conditioned semantic distributions. Additionally, we compute FID scores and have reported them below  with respect to the set of neutral images. The FID value is lowest for American nationality (in agreement with our TDS and WTD), while the highest is for Iranian nationality, closely followed by Egyptian and Nigerian nationalities. We believe that including semantics is a better metric, and hence we reiterate that the proposed metrics provide a more accurate representation.
>
> | Nationality | FID (from Neutral) |
> |---|---|
> | American | 52.38 |
> | Chinese | 84.75 |
> | Egyptian | 91.05 |
> | French | 61.33 |
> | Indian | 79.42 |
> | Iranian | 96.90 |
> | Japanese | 92.13 |
> | Mexican | 88.67 |
> | Nigerian | 89.68 |
> | Russian | 69.19 |
>
> We have also added a subsection (Section 5.1) in our supplementary regarding the computation of FID scores and the observations.
>
> **4. Neutral baseline potentially encoding Western/majority norms:** We believe that neutral baselines are encoding norms for the most represented group in the training dataset. Moreover, we do not claim that a nationality group closer to the neutral baseline is bias-free. Instead, we observed that highly represented groups have higher diversity (as seen in t-SNE visualization as well). We do not claim that the perfect model is one that can generate non-distinct images; instead, we claim that the complete pipeline from generation to multimodal inference can generate stereotypes. We have revised the paper to include stronger language indicating this intention, added a paragraph on the neutral baseline in Section 3.1 of the main paper, and included a separate t-SNE visualization for each T2I model, showing how the different distributions are mixed and used for analysis.
>
> **5. Stereotype vs. recognition (RQ2):** We agree that some inferred traits may be entangled with contextual cues (e.g., attire, setting, co-occurring objects), the goal is not to "prove intent" or to treat every deviation as a stereotype, but to audit how text-to-image generations couple sensitive demographic mentions with downstream social inferences. For instance, SORA frequently generates French men holding bread, leading MLLMs to assign "chef" - a contextually plausible but demographically patterned inference. Across these settings, we consistently observe that demographic mentions systematically bias the distribution of predicted occupations/traits beyond what is explained by purely visual recognition, precisely the operational risk we aim to quantify. We now also explicitly state in the limitations that context is a mediator and that our analysis should be interpreted as measuring associative bias in generated content, not ground-truth demographic facts or causal intent.

---

> > ### Author Response · Authors · 2026-01-19
> > **Response for Reviewer aBQz (Contd.)**
> >
> > **6. Stereotype-specific behavior (corruption instability) vs. generic robustness:** We agree that corruptions can induce generic prediction instability, and we do not claim otherwise. Instead, our hypothesis (and revised claim) is narrower: corruptions disproportionately amplify instability in socially sensitive, high-consequence label spaces, even when the underlying image content remains semantically similar. To support this, we introduce a "meaningful change" criterion and a threshold sensitivity analysis for MCS, and we replicate the behaviour on real photographs (UTKFace) in addition to generated images. Importantly, the observed shifts under corruption are not merely random noise; they show a consistent tendency toward semantically related but socially charged descriptors (e.g., "teacher" to "criminal" rather than arbitrary shifts to, as shown in Figure 5b), which creates a practical deployment risk: minor distribution shifts can flip a system's socially sensitive interpretations more than expected from generic robustness alone. We therefore position RQ3 as an auditing lens for robustness of social inferences (not as a general robustness benchmark), and we include this scope clarification explicitly in the revised paper.
> >
> > **7. Additional Comments:** We have addressed the additional comments by relocating the footnote disclaimer from Page 2 to Page 1. As an alternative to treemaps, pie charts have been added to the Supplementary material.

---

### Review · Reviewer_R8R3 · 2026-01-05

**Summary Of Contributions:**

This paper introduces an auditing framework for analyzing stereotypical biases in T2I generative models and MLLMs across different nationalities and genders. The framework addresses three research questions: **(i)** how stereotypes manifest in generated images, **(ii)** how MLLMs infer occupations and interests from generated images across demographic groups, and **(iii)** how natural image corruptions affect the stability of these biases.

The authors propose several quantitative metrics (e.g., TDS, WTD, SDI, OM, and MCS) to summarize bias patterns and apply them to recent models such as SDXL, LLaVA, and Qwen-2.5-VL. The study combines embedding-based analyses, frequency-based probing, and qualitative visualizations to provide a systematic view of demographic stereotypes in multimodal generative systems.

Strengths

- The paper investigates stereotypical biases on both T2I and MLLM models

- It unifies the metrics available for stereotypical biases

- The use of treemaps and before–and–after qualitative examples provides an interpretable view of stereotypical patterns under image degradation.

- It evaluates recent, practically relevant models and considers both open- and closed-vocabulary settings.


Weaknesses

- Several proposed metrics are closely related to existing distributional or frequency-based measures used in prior bias auditing work, which limits methodological novelty.

- The notion of “meaningful change” under degradation depends on heuristic thresholds, which may affect the interpretability and robustness of conclusions.

**Audience:**

Yes

**Audience Explanation:**

The findings are likely to be of interest to researchers and practitioners working on multimodal learning and AI fairness. In particular, the evaluation of T2I models and MLLMs reflects realistic deployment pipelines, and the analysis of bias stability under image corruptions is relevant for understanding robustness in real-world settings. While the methodological contributions are incremental, the paper provides a consolidated and up-to-date empirical perspective that can serve as a useful reference point for future work on bias evaluation and mitigation.

**Broader Impact Concerns:**

The paper includes an appropriate Broader Impact Statement that clearly positions the work as an auditing effort for identifying and understanding stereotypical biases in multimodal models. The main ethical consideration that could be further clarified is the use of highly stigmatizing labels in the predefined occupation set; while motivated as a stress test, additional guidance on responsible interpretation would strengthen the ethical framing. Otherwise, no significant unaddressed broader impact concerns are identified.

**Claims And Evidence:**

Yes

**Claims Explanation:**

The experimental results presented generally support the claims made in the paper. The authors conduct experiments across multiple T2I and MLLM models, demographic groups, and evaluation settings, and the reported results consistently demonstrate stereotypical patterns aligned with the stated research questions. The use of embedding visualizations, trait deviation statistics, frequency analyses, and qualitative examples provides converging evidence for the presence and persistence of biases.

However, while the evidence is clear and reproducible, it is largely observational. The experiments confirm that biases exist and can persist under image degradation, but they do not fully establish how specific model components contribute to these effects. As a result, the claims are well-supported empirically, though primarily at a descriptive rather than explanatory level.

**Requested Changes:**

Clarify the relationship to prior metrics and frameworks. The paper would benefit from a more explicit discussion of how the proposed metrics relate to existing divergence-, entropy-, or frequency-based bias measures, and what practical advantages they offer beyond prior work.

Strengthen the analysis of image degradation effects. Providing additional justification for the MCS thresholding strategy or a sensitivity analysis would improve confidence in the conclusions regarding bias stability.

---

> ### Author Response · Authors · 2026-01-19
> **Response for Reviewer R8R3**
>
> We thank Reviewer R8R3 for their time and review. Regarding the descriptive nature of our work, we would like to clarify that the framework is intended as a diagnostic auditing tool, identifying where and under what conditions stereotypes emerge, rather than attributing them to specific internal components.
>
> **1. Relationship of proposed metrics to prior work:** TDS and WTD are related to distributional divergence measures, but are defined over the attribute-conditioned trait distribution relative to a neutral baseline. Further, SDI is connected to a frequency-based dominance measure, but it is normalized against the global distribution for cross-group and cross-model comparisons. OM is based on an overlap-based similarity measure and also accounts for shared stereotypical labels across demographic groups, unlike prior divergence-based metrics. These metrics are meant to be interpretable and directly actionable for mitigation strategies. In addition, we have added the table below to the supplementary material (as Table 3) to compare related metrics.
>
> | Metrics | Closest Prior | Key Differences |
> |---|---|---|
> | TDS | Distributional Divergence | With the demographic conditioning, it computes dominant traits relative to a neutral baseline. |
> | WTD | KL Divergence | Deviations captured across multi-trait stereotype patterns that are extendable across the board. |
> | SDI | Frequency Dominance | Normalized against the global label distribution for cross-group and cross-model comparison. |
> | OM | Jaccard Overlap| Reveal common stereotypes between different groups. |
> | MCS | Agreement/ Stability | A semantic-based computation for uncertain but realistic conditions. |
>
>
> **2. Heuristic thresholding in Meaningful Change Score (MCS):** We agree that robustness should be demonstrated, and as such, we have performed a sensitivity analysis using different thresholds. We have reported the results in the table below and in Table 1 of the supplementary material. We observe that the MCS value increases monotonically.
>
> | Nationality | δ=0.50 | δ=0.55 | δ=0.60 | δ=0.65 | δ=0.70 |
> |---|---|---|---|---|---|
> | American | 0.189 | 0.197 | 0.208 | 0.208 | 0.213 |
> | Chinese | 0.179 | 0.183 | 0.188 | 0.189 |  0.206 |
> | Egyptian | 0.178 | 0.186 | 0.194 | 0.195 | 0.213 |
> | French | 0.174 | 0.178 | 0.185 | 0.185 |  0.196 |
> | Mexican | 0.215 | 0.217 | 0.222 | 0.225 | 0.249 |
> | Indian | 0.151 | 0.151 | 0.152 | 0.156 | 0.181 |
> | Iranian | 0.165 | 0.17 | 0.171 | 0.173 | 0.191 |
> | Nigerian | 0.211 | 0.223 | 0.231 | 0.233 | 0.247 |
> | Japanese | 0.207 | 0.214 | 0.218 | 0.219 | 0.242 |
> | Russian | 0.202 | 0.209 | 0.212 | 0.214 | 0.23 |

---

### Decision · Action_Editor_PWmV · 2026-02-19

**Recommendation:** Reject

**Audience:**

Yes

**Audience Explanation:**

The question of how stereotypes propagate across text-to-image generation and downstream multimodal interpretation is relevant to researchers working on generative modeling and fairness in machine learning and generative AI. The paper examines this issue in an end-to-end setting and reports empirical patterns related to bias evaluation and robustness in such pipelines. These topics fall within the scope of ongoing research discussions in the machine learning community.

**Claims And Evidence:**

No

**Claims Explanation:**

The paper presents an interesting study of stereotypes in text-to-image models and MLLM pipelines. It includes a fairly broad empirical analysis, and the proposed metrics seem helpful for structuring and visualizing the results. However, given the reviewers’ assessment, I do not find the evidence convincing enough to support the claims discussed in the manuscript.

As noted by the reviewers, an important concern is that the authors’ conclusions about model behavior rely heavily on embedding-based measurements, without numerical validation that these measurements reflect the ground truth or are independent of the chosen embeddings. This includes the paper’s conclusions about which nationality groups are more or less stereotyped and how physical-trait deviations are interpreted. Specifically, these findings are derived from CLIP or SigLIP embedding similarity and embedding-space analyses. However, the paper does not demonstrate that these embedding-based proxies recover ground-truth traits verified through human subject studies or controlled verification experiments in settings with known or pre-defined ground truth. Averaging across only two embeddings provides a limited robustness check and does not address potential shared biases in the evaluation layer itself.

Furthermore, the metric design introduces degrees of freedom that can significantly influence the reported outcomes. For example, the WTD metric depends on the choice of divergence, and the paper uses the $L_1$ distance without demonstrating robustness of the conclusions to alternative divergences. This leaves open whether the reported rankings and deviation magnitudes are intrinsic to the data or partly driven by metric design choices. Similarly, as noted by Reviewer aBQz, the interpretation of “distance from neutral” is complicated by the fact that the neutral baseline reflects the model’s default prior, which may itself be biased.

Given these comments and weaknesses, the empirical findings and conclusions could be potentially shaped by the choice of embeddings and metric definitions rather than reflecting the models' ground-truth features. The manuscript does not provide sufficient numerical evidence that the proposed measurement pipeline reliably captures stereotype severity independent of embedding and metric design choices.

**Resubmission Of Major Revision:**

The authors may consider submitting a major revision at a later time.